# INSUFFICIENT TASK DESCRIPTION CAN IMPAIR IN-CONTEXT LEARNING: A STUDY FROM INFORMATION PERSPECTIVE

## ABSTRACT

Transformers have demonstrated remarkable performance in a wide range of applications, making in-context learning an essential technique. In-context learning primarily relies on two types of information: in-context examples and task description. While previous research has extensively investigated the influence of in-context examples on learning behavior, the role of task description has not been adequately explored, despite their practical significance. In this paper, we present a study examining the impact of task description on the in-context learning performance of transformers. We devise a synthetic experiment setting, making the information of task description controllable. Through a series of well-designed experiments, we systematically vary task description information and assess the resulting effects on model performance across multiple tasks. Our findings reveal the double-side roles of task description: insufficient task description will lead the model to ignore in-context examples, resulting a poor in-context performance; once the information in task description surpasses a certain threshold, the impact of task description transfers from negative to positive, and a performance emergence can be observed. We further conduct the tasks on GPT-4 and observe a similar double-side impact. In conclusion, this study contributes to a deeper understanding of the in-context learning from a task description perspective.

## 1 INTRODUCTION

In-context learning refers to the transformer's ability to learn from context-based prompts, which has been utilized in numerous applications, including AI planning (Valmeekam et al., 2022; Xie et al., 2023), reasoning (Huang & Chang, 2022), image understanding (Alayrac et al., 2022) and autonomous agents (Wang et al., 2023). Despite the extensive application of in-context learning, our comprehension of its underlying mechanisms is still underexplored. Recent research has investigated in-context learning within a meta-learning framework (Gu et al., 2023; Min et al., 2021), offering insights into how transformers utilize in-context examples to tackle new tasks. However, transformers can employ in-context information from two sources: in-context demonstrations and task description. The role of task description, though practically significant, has not been thoroughly studied. In this work, we concentrate on how task description influences in-context learning within a meta-learning framework.

The meta-learning framework (Gu et al., 2023; Min et al., 2021) is used to enrich in-context learning of transformer, where the transformer is directly trained to implement in-context learning. We adopt an arithmetical operation, which has been widely utilized to study in-context learning (Akyürek et al., 2022; Power et al., 2022; Garg et al., 2022; Razeghi et al., 2022). Specifically, each task can be constructed as $((a \cdot x) \circ (b \cdot y)) \bmod p = r$, where $x, y$ are the inputs, $p$ is a prime number, $\circ$ represents an operator, and $r$ is the result to be predicted. $a, b, \circ$ together specify the task. The transformer is expected to learn this task from the few shot examples, and the prompt is formulated as $[\{(x_i, y_i, r_i)\}_{i=1}^{l}, (x_q, y_q)]$. $\{(x_i, y_i, r_i)\}_{i=1}^{l}$ can be regarded as few shot examples, while $x_q, y_q$ is the query.

In this paper, we aim to study the impact of task description. Unlike the previous setting, we include the task description in the prompt. Specifically, the prompt in our modified framework is

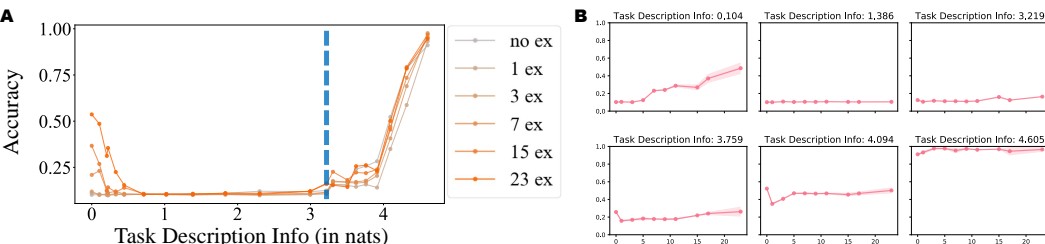

Figure 1: **A:** A transition of the impact on the performance of in-context learning can be observed given different amount of task description information. Beyond 3.2 nats (marked with a blue dashed line), the task description acts as a positive role and boosts performance significantly. Before the threshold, the information of task description has little (in the range of 0.8~3.2 nats) or even negative (in the range of 0~0.8 nats) impact ( lower than task info = 0.). **B:** Influence of in-context examples (x-axis) on in-context learning performance (y-axis), given different task description. Shaded areas indicates +/- std calculated from 3 runs. Task description info is measured in nats.

$[d, \{(x_i, y_i, r_i)\}_{i=1}^{l}, (x_q, y_q)]$, where $d$ denotes the task description depicting $a, b, \circ$. To investigate the role of task description, we devise a synthetic experiment, where we can flexibly control the complexity of the task description $d$, i.e., assigning $d$ with different levels of information of $a, b, \circ$. Specifically, given a task ground truth label $t = (a, b, \circ)$, we design task description $d$ to control the mutual information $I(t; d)$.

In the proposed experimental setup, we investigate the impact of task description on in-context learning capability when changing the mutual information $I(t; d)$.

As shown in Figure 1A, we observe a transition regarding the impact of task description: those with insufficient information can impair in-context learning, while task description with abundant information can aid it, with a transition between these states. We identify two cases **where negative impacts of in-sufficient task description on in-context learning can be somewhat mitigated**: (i) a large number of in-context examples with low-information task description, and (ii) high-information task description. We also investigate whether incorporating task prediction as an auxiliary task improves in-context learning. Results indicate that task prediction as a surrogate task generally benefits in-context learning. To verify the generality of our findings, we conduct further studies on more realistic NLP tasks, the results also align with our experimental results on the synthetic tasks. We further let GPT-4 to perform our synthetic tasks with task description, observing similar trends in predictions. These experiments confirm the universality of our findings. In summary, we study in-context learning from the perspective of task description, and reveal that task description with insufficient information can have a negative impact on in-context learning.

## 2 RELATED WORK

**In-context learning**  Recent years have seen significant advances in natural language processing (NLP), especially with the development of large-scale language models designed for in-context learning. These models, such as GPT-4 (OpenAI, 2023) by OpenAI, PaLM2 (Anil et al., 2023) by Google, and Llama (Touvron et al., 2023) by Facebook, excel in understanding and generating human-like text by leveraging massive amounts of data and sophisticated algorithms. In-context learning refers to the model's ability to adapt its understanding and responses based on the specific context provided (Brown et al., 2020), which has been proven to be crucial in enhancing their performance across various NLP tasks, including AI planning (Valmeekam et al., 2022; Xie et al., 2023), reasoning (Huang & Chang, 2022), image understanding (Alayrac et al., 2022), and autonomous agents (Wang et al., 2023). However, despite the impressive progress, challenges persist in understanding the mechanism behind in-context learning. This paper delves into this mechanism through synthetic tasks, taking a step forward in grasping in-context learning from the aspect of task description.

**Exploration of in-context learning from synthetic tasks.**  Exploring in-context learning mechanisms in real-world applications is challenging due to their complexities (Min et al., 2022). Recent

studies have turned to synthetic tasks for a more controlled environment to understand these mechanisms effectively. For instance, linear regression tasks have been used in several studies (Akyürek et al., 2022; Von Oswald et al., 2023; Garg et al., 2022) to explore Transformer's in-context learning behavior, while some researchers also study image data to analyze the learning process. Moreover, investigations (Chan et al., 2022a;b; Fu et al., 2023) have been conducted from in-context and in-weights perspectives, examining the learning process through the lens of the model's internal representations and the role of weights. While valuable, many of the aforementioned explorations tend to overlook the impact of task descriptions on the in-context learning process. Understanding this influence is crucial for guiding these models toward desired learning outcomes and enhancing their effectiveness across various applications.

**Task description in real in-context learning application.** In the realm of in-context learning, the prompt plays a crucial role in guiding the language model's response generation. A prompt is a textual input provided to the model, containing the necessary context and instructions that help the model understand the user's requirements and produce relevant responses. The task description in the prompt often includes specific questions, statements, or examples that outline the desired output, enabling the model to adapt and generate contextually appropriate text (Brown et al., 2020). The task description plays a important role in in-context learning by providing information about recognizing the task in real application (Pan, 2023; Cho et al., 2023). However, systematic studies about the role of task description and the mechanisms behind are lacking. This paper, from a information perspective, fills this gap by providing the analysis of task description under different situations. Our work can provide a general guidance on how to providing task description.

## 3 FORMULATION AND MOTIVATION

We assume a dataset $\mathcal{D}$, comprising $N$ data samples $\mathcal{D} = \{(d_j, c_j, q_j, r_j, t_j)\}_{i=1}^N$, where the $j$-th sample is a specific task, containing the task description $d_j$, a sequence of task examples $c_j = \{(x_i, y_i, r_i)\}_{i=1}^l$, a query $q_j$ and related output $r_j$. $t_j$ contains the elements specifying the task, in our synthetic task, $t_j = (a_j, b_j, \circ_j)$. The examples satisfy $((a_j \cdot x_i) \circ_j (b_j \cdot y_i)) \bmod p = r_i$ We partition the dataset into two subsets: $\mathcal{D}_{train}$ and $\mathcal{D}_{test}$. This partitioning should ensure that tasks in the test dataset remain unseen in the training dataset, i.e., for each task $k$ in the testing set $\mathcal{D}_{train}$, no $t_j$ exists in $\mathcal{D}_{test}$ such that $t_k = t_j$. The primary aim of in-context learning is to utilize the task description and examples to adapt the model, thereby optimizing its performance on previously unseen tasks. To accomplish this objective, we maximize the following function:

$$\mathbb{E}_{p(d,c,q)}\mathbb{E}_{q_\theta(r|d,c,q)} \log p(r|d,c,q). \tag{1}$$

Here $q_\theta(r|d,c,q)$ denotes the predicted distribution of target $r$, while $p$ refers to real distribution. To analyze the aforementioned objective associated with task label $t$, we employ the variational method, constructing an evidence lower bound. Given the intractable nature of the distribution $p(t|r,d,c,q)$, we approximate it using a parameterized distribution $q_\theta(t|d,c,q)$ as follows:

$$\begin{aligned}
&\mathbb{KL}(q_\theta(t|d,c,q)|p(t|r,d,c,q)) \\
&= \mathbb{KL}(q_\theta(t|d,c,q)|p(t|d,c,q)) - \mathbb{E}_{q_\theta(t|d,c,q)} \log p(r|t,d,c,q) + \log p(r|d,c,q).
\end{aligned} \tag{2}$$

Please refer to appendix B for the proof. Considering the non-negative nature of the KL divergence, we can express the log-likelihood in the following manner:

$$\log p(r|d,c,q) \geq -\mathbb{KL}(q_\theta(t|d,c,q)|p(t|d,c,q)) + \mathbb{E}_{q_\theta(t|d,c,q)} \log p(r|t,d,c,q). \tag{3}$$

The first term signifies the task label prediction, whereas the subsequent term corresponds to the loss function employed in the in-context training for the transformer. This equation, therefore, demonstrates that accurate task label prediction contributes to the maximization of the log-likelihood.

Incorporating the task description as a component of the input allows it to serve as a representation of the task itself. To assess the efficacy of this description, we examine encoder and decoder models that yield conditional distributions $q(d|t)$ and $p(t|d)$. Given that $q(t)$ embodies the marginal distribution of task $t$, we define the reconstruction error, denoted as $\mathcal{R}$, in the following manner:

$$\mathcal{R} = \mathbb{E}_{q(t)}\mathbb{E}_{q(d|t)}[-\log p(t|d)] \leq \mathbb{KL}(q(t,d)||p(t,d)) - I(t;d) + H_q(t), \tag{4}$$

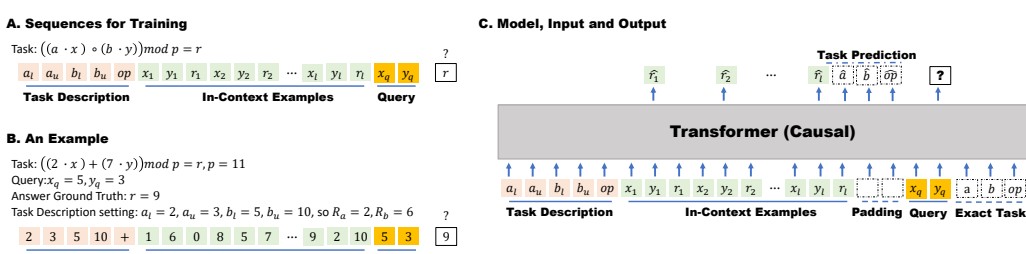

Figure 2: Experimental Setup. **A:** We construct our synthetic task dataset using basic equations. During training, the model receives a prompt sequence with task description, in-context examples, and a query. Only an inexact range of $a$ and $b$ will be implied in task description, and we train the transformer to calculate the answer $r$ of the operation given $x_q$ and $y_q$ as query. **B:** An example of prompt sequence. $R_a = a_u - a_l + 1$, $R_b = b_u - b_l + 1$, and $a_l, a_u, b_l, b_u$ stand for the possible lower and upper bounds of $a$ and $b$. **C:** Model, input and output. We use standard decoder-only Transformer, taking a token sequence as input and employing auto-regression for training. We calculate loss for the output sequence, and accuracy is measured solely on the answer of query equation. For task prediction, exact task label $t = (a, b, op)$ will be added to the end of input token.

where $I(t; d)$ is the mutual information between task label $t$ and the task description $d$. Please see appendix C for the proof. The aforementioned equation indicates that increasing the mutual information can reduce the negative log likelihood of $t$.

Further, the mutual information $I(t; d)$ can be bounded as follows:

$$0 \leq I(t; d) = \mathbb{E}_{p(t,d)} \left[ \log \frac{q(t, d)}{q(t)q(d)} \right] = H_q(t) - H_q(t|d) \leq H_q(t). \tag{5}$$

Based on the aforementioned equation, we observe that the mutual information ranges from 0 to $H_q(t)$. Consequently, to examine the impact of mutual information, we propose incorporating its control in our experimental design. Please see Sec. 4 for details.

In summary, we consider an in-context learning setting where the task is unseen in the training set. However, to simplify the problem, we assume that the task labels in the testing set are novel recombinations of the training ones. In order to reformulate the prediction into a compositional generalization problem, we derive a variational lower bound of the log-likelihood as a new objective, as shown in Equation 3. The first term in it is for task prediction. Since we consider the task description as a representation of the task, its goodness has an impact on the model performance. By modeling it as a representation, we derive a quantity to estimate its goodness, as shown in Equation 4. Therefore, we design our experiments with some principles to analyze how to train our model for better in-context ability from the following perspectives: $(i)$ **the mutual information between the task description and the task label**; $(ii)$ **with or without task prediction**.

## 4 EXPERIMENTAL DESIGN

In this section, we delve into the experimental design to conduct the study on task description from the above two perspectives. We begin by outlining the design principles, which serve as the foundation for the entire experiment.

**Design Principle** $(i)$ **Controllable task description information**: The information provided in the task description can be directly manipulated, allowing for precise control over the quantity of information presented to the model. $(ii)$ **Unseen evaluation tasks**: To ensure the model's ability to generalize, the evaluation tasks presented to the model are not included in the training set, which is necessary to assess the model's performance on unseen tasks. $(iii)$ **Information inference from multiple sources**: The model is designed to extract information from both the task description and in-context examples provided. This enables the model to adapt and learn from various sources of information.

## 4.1 TASK DESIGN WITH CONTROLLABLE TASK DESCRIPTION

As mentioned before, each datum in our synthetic dataset is constructed as $x = (d, c, q, r, t)$, we remove the index for simplicity. $t = (a, b, \circ)$ is the ground truth label of the equation. $c = \{(x_i, y_i, r_i)\}_{i=1}^l$, and each item satisfies $((a \cdot x_i) \circ (b \cdot y_i)) \mod p = r_i$. For $\circ$, we choose $+, -$ or $/$. $q$ and $r$ are query and related output.

As shown in Figure 2AB, we train the transformer to predict the output $r$ of query $q$, given a task description $d$ and in-context examples $c$. The prompt is formulated as $[d, c, q]$.

Here we provide the details of task description $d$. Following the design principle above, we set the task description $d$ to provide the range of $a, b$. Specifically, the task description is given as $d = (a_l, a_u, b_l, b_u, op)$, and $a \in [a_l, a_u]$ and $b \in [b_l, b_u]$. $op$ stands for the operator $+, -$ or $/$. We provide definite information of the operator, and control varying mutual information between $d$ and $t$ by using different $a_l, a_u, b_l, b_u$. Specifically, the total number of possible $a, b$ pairs is $n_{ab}$. For example, $n_{ab} = 100$ when $a, b \in [1, 10]$. Given this task description, we can narrow down possible $ab$ pair numbers from $n_{ab}$ to $R_a \cdot R_b$, where $R_a = a_u - a_l + 1$ and $R_b = b_u - b_l + 1$. This indicates that the information given by the task description can be formulated as:

$$I(t; d) = log(\frac{n_{ab}}{R_a \cdot R_b}). \tag{6}$$

We choose $p = 11$ in all experiments. $a, b$ are integers, and $a, b \in [1, 10]$. We set the above range of $a, b$ for efficiency, which can already demonstrate the impact transition of task description. Finer-grained control of the task description information can be achieved with larger range of $a, b$. When setting $R_a, R_b$ to the largest range, $n_{ab} = R_a \cdot R_b$, and $I(t; d) = 0$, zero task information is provided, which we treat as the baseline. The task description tokens shown in Figure 2AB are still provided for fair comparison.

## 4.2 MODEL TRAINING WITH TASK PREDICTION

**Loss Function** The auto-regression is used to train the model. Following GPT (Radford & Narasimhan, 2018), given a token sequence $z = (z_1, \ldots, z_T)$, we train the model to predict $p(z) = \prod_{t=1}^T p(z_t | z_{<t})$. We calculate loss for in-context examples, query, and the answer of query equation. The in-context examples are denoted as set $\mathcal{C}_{i-1}$. For $i > 1$, $\mathcal{C}_{i-1}$ represents the in-context example sequence $\{(x_1, y_1, r_1), \ldots, (x_{i-1}, y_{i-1}, r_{i-1})\}$. For $i = 1$, $\mathcal{C}_0$ is an empty set. Specifically, we calculate the loss for the sequence $s = \{(x_1, y_1, r_1), \ldots, (x_L, y_L, r_L)\}$ and task description $d$ as follows:

$$\mathcal{L}(\theta, s, d) = \frac{1}{L} \sum_{i=1}^L l(f(\{d, \mathcal{C}_{i-1}, x_i, y_i\}), r_i), \tag{7}$$

where $l$ denotes the loss function, e.g., cross entropy loss is adopted in our setting. Accuracy is calculated only for the answer $r$ of query equation.

For task prediction, as shown in Figure 2C, task label $t = (a, b, op)$ will be added to the end of input token, to add in task prediction loss while avoiding task information leakage (as auto-regression ensures the model outputs the predicted answer before seeing the task label). Loss for task prediction can be re-formulated as:

$$\mathcal{L}_t(\theta, s, d) = \frac{1}{L} \sum_{i=1}^L l(f(\{d, \mathcal{C}_{i-1}, x_i, y_i\}), r_i, t). \tag{8}$$

**Model and Training** For experiments on synthetic tasks, we use a standard decoder-only causal Transformer (Vaswani et al., 2017) with 24 layers. For experiments on the natural language task CoFE (An et al., 2023), we follow their approach and use pre-trained GPT2-Large as our model. We use Adam optimizer with learning rate $1e^{-4}$ for all experiments. To reduce randomness, we calculate the mean value of 3 runs. More implementation details are given in appendix A.

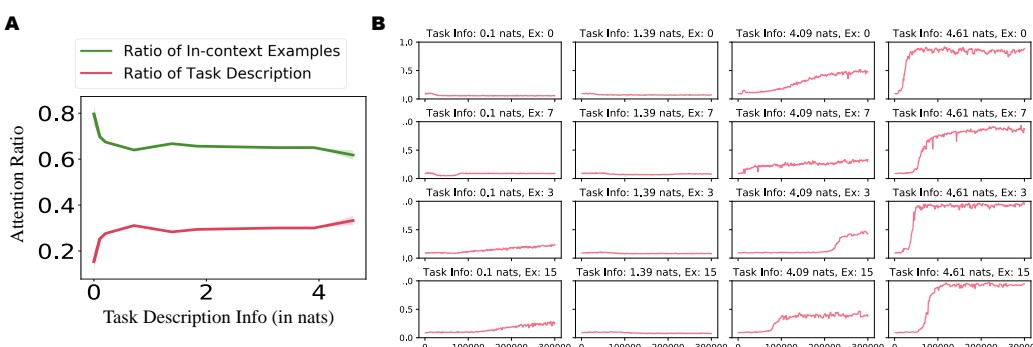

Figure 3: Exploring the transition of task description. **A:**Attention explanation for interference between task description and in-context examples in transformer's in-context learning . As task description info increases, the attention ratio for examples decreases. For all experiments, 15 in-context examples are given. **B:** The transformer's learning curve, showing the transformer's accuracy on validation queries (y-axis) across training steps (x-axis). Notable in-context learning process is evident with sufficient task info or when in-context examples' impact surpasses incomplete task description's distraction (low task info, many examples).

## 5 RESULTS

### 5.1 HOW TASK DESCRIPTION AFFECTS IN-CONTEXT LEARNING

**Negative impact of Insufficient Task description Can be Observed.** We use the accuracy of the predicted results $r$ of queries $(x_q, y_q)$ to reflect in-context learning performance, and use the mean of three runs to reduce the randomness. The results are presented in Figure 1A. Before a certain information threshold (about 3.2 nats), the task description has negative ($0 \sim 0.8$ nats) or little ($0.8 \sim 3.2$ nats) impact on the in-context performance. Specifically, for $0 \sim 0.8$ nats, more task information leads to worse performance, especially for relative large number of in-context examples, such as 15, 23. For $0.8 \sim 3.2$ nats, the accuracy remains at a low level. Interestingly, after the information threshold, the accuracy grows rapidly as the information gains, but keeps relatively stable with changes in the number of in-context examples.

In general, there are two cases where the model can achieve a relatively high accuracy: $(i)$ given low-information task description, and large number of in-context examples, or $(ii)$ given high-information task description.

We try to understand the observed transition by analyzing how accuracy relates to task description information and the number of in-context examples. We select 6 representative points: 2 points (0.104, 1.386) before the transition, 1 point around the transition (3.219), and 3 points after the transition (3.759, 4.094, 4.605), as shown in Figure 1B. When task information is at low level (0.104), in-context examples dominate in transformer's learning and the accuracy grows as the number of in-context examples increases. When task description is sufficient (4.605), the performance is high and in-context examples have minimal impact.

When task information is insufficient (3.759, 4.094), adding one in-context example slightly decreases accuracy compared to no in-context examples, suggesting distraction from the added example. On the other hand, when **medium-level task information** added (1.386, 3.219), the accuracy do not increase given more in-context examples and the accuracy is lower than 0.104, indicating the added task information actually misleads the learning on the in-context examples. In a summary, as there are two sources, the task description and in-context examples, providing the task information, the model may struggle on relying which source to capture the in-context ability, i.e., the task information and in-context examples can interfere with each other, leading to poor in-context capability.

To verify the above hypothesis, we further analyse the attention inside the transformer. Given same input sequence, we sum up weights in all attention layers in the transformer, and calculate the ratio of in-context examples and task description respectively. The number of in-context examples is fixed to 15. As shown in Figure 3A, the ratio of in-context examples in attention keeps declining with more

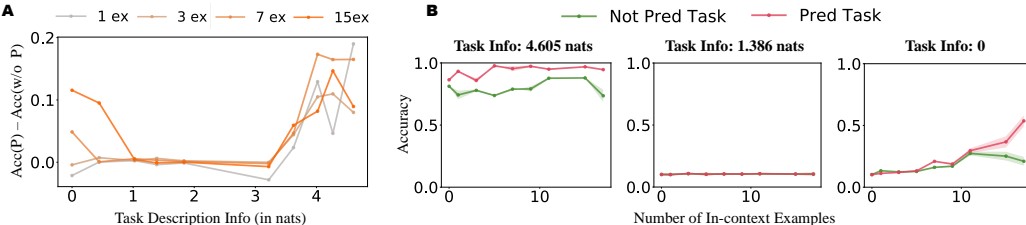

Figure 4: Results of task prediction. **A:** A demonstration of accuracy gain (Predicting tasks v.s. without predicting tasks). Acc(P) refers to accuracy on predicting results $r$ under predicting task setting, Acc(w/o P) refers to corresponding accuracy without task prediction. Accuracy gain means the value of Acc(P) - Acc(w/o P). **B:** Comparison between Acc(P) and Acc(w/o P) given various amount of task information and various number of in-context examples.

task description information. On the contrary, the attention ratio of task description increases when more task-related information are given. This indicates that adding task description info will divert model's attention in in-context examples.

We also demonstrate the accuracy during the training process. Figure 3B illustrates how the transformer struggles to learn with incomplete task details (before the transition threshold 3.2 nats). Only after the transition, an obvious in-context learning process can be witnessed given sufficient task info. Or the transformer can learn gradually when given very low task info and large number of in-context examples (refers to the first column in Figure 3B), mainly because the influence of in-context examples can exceed the distraction of incomplete task description. In appendix F, we attempt to provide a rough explanation to above phenomenon using a simplified 1-layer transformer's attention mechanism.

**Higher information of task description will increase the performance.**    As illustrated in Equation 4, higher mutual information will reduce the upper bound of the prediction error. Intuitively, the task description captures the essential aspects and the underlying structure of the task, providing the model with valuable insights and a more accurate understanding of the problem it needs to solve. When the mutual information is high, it means that knowing the task description reduces the uncertainty about the prediction of task itself. Consequently, when the task description has high mutual information with the task, the model can leverage this strong representation to make better decisions and predictions, even when faced with limited or ambiguous examples.

### 5.2    IMPACT OF TASK PREDICTION

To study how predicting task description affects in-context learning performance (measured through the accuracy of the predicted results $r$ of validation queries), we conduct experiments by adding an extra loss between the predicted task label and ground truth task label. By comparing the gain (with predicting task label v.s w/o predicting task label), we can evaluate the impact of task prediction.

**Predicting the task can improve in-context learning performance.**    We calculate the accuracy (with predicting task description) minus the accuracy (w/o predicting task description), and the results are presented in Figure 4A. A performance improvement can be observed under different task descriptions and in-context example settings, as the curves mostly stay above zero axis. Figure 4B provides a more detailed comparison under different levels of task information: high (4.605 nats, after the Transition), medium (1.386 nats, causing distraction in transformer's in-context learning) and very low (0 nat, no distraction). These results confirm that predicting the task label generally enhances in-context learning performance, except when the transformer's in-context learning ability is distracted by incomplete task info.

### 5.3    PERFORMANCE OF GPT-4 ON THE SYNTHETIC TASK

**Synthetic experiments using GPT-4 result in similar performance pattern.**    We test if LLMs are affected similarly by task descriptions by replicating the synthetic experiment with GPT-4 (OpenAI,

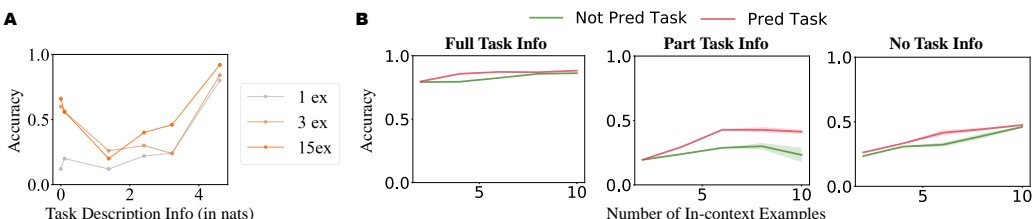

Figure 5: Additional experiments. **A**: Performance of GPT4 on our synthetic task. Incomplete task description can cause similar distraction in GPT4 in-context learning ability. **B**: Experiments on real tasks. We design three different settings of task description, and experiment on all three info settings given 2, 4, 6, 8, 10 in-context examples separately.

2023). We sample task description and corresponding in-context examples from the synthetic task validation dataset to form prompt for GPT-4, and require GPT-4 to predict the possible real task and result to the queries. For each experiment setting (given certain amount of task info and certain number of in-context examples), we randomly sample 50 prompts and the resulting accuracy of the predicted results of queries is presented in Figure 5A. Similar performance pattern appears when given incomplete task description (around 2 nats), that GPT-4 performs worse than given less task info (around 0 nats) due to interference between task description and in-context examples. Only when the in-context examples are insufficient for in-context learning (1 example) does accuracy improve with more task info.

### 5.4 Beyond the Synthetic Experiment

To verify that the discovery from the synthetic experiment also hold on the real task, we conduct another experiment on the more realistic task on several NLP tasks. The results are well-aligned with the findings on the synthetic experiment, indicating our findings can be well scale to real word cases.

**Experiments on compositional generalization tasks.** We experiment on CoFE (An et al., 2023), a natural language dataset focused on compositional generalization, which assesses a model's ability to generalize and predict novel combinations based on its training data.

In our experiments, we select 12 task labels from CoFE, with the training set comprising 4 randomly chosen labels and the test set containing the remaining 8. More details and examples of CoFE data are provided in appendix D.

We design three settings of task description containing different amount of information: Full Task Info includes all task information, Part Task Info implies only the task label category, and No Task Info omits task description entirely. Across these settings, we vary the number of in-context examples.

Results in Figure 5B indicate that using task prediction as a proxy task can still significantly improve accuracy, highlighting its impact on the transformer's in-context learning. Full Task Info consistently achieves the highest accuracy, suggesting that detailed task information enhances in-context learning ability. Conversely, incomplete task descriptions lead to lower accuracy and limited gains with more in-context examples.

**Experiments on spelling tasks.** We perform further validation on spelling tasks introduced in (Honovich et al., 2022). The tasks are delineated in Table 1. We use instruction as task description. For both tasks, we experimented on three scenarios: full instruction, in-complete instruction and no instruction. All Experiments based on pre-trained GPT-4.

Table 1: The instruction of spelling tasks.

| Task | Instruction | example |
|---|---|---|
| Second Letter | Extract the second letter of the input word. | input: cat, output: a |
| Starting With | Extract the words starting with a given letter from the input sentence. | input: Has the potion not worked? [p], output: potion |

For "Second Letter" task, we use "Extract letter of the input word" as in-complete task description, and evaluate the model's ability to learn in-context information with varying numbers of examples. For 'Starting With' task, we use 'Extract words from the input sentence' as in-complete task description. Resulting accuracy given in Table 2. Similarly, model given no task description outperforms model given in-complete task description (when given 10 examples).

Table 2: The results of spelling tasks.

| Task | No Task Info | Part Task Info | Full Task Info |
|---|---|---|---|
| Second Letter | 0.5 | 0.1 | 1 |
| Starting With | 0.85 | 0.45 | 0.95 |

### 5.5 Ablations

**Experiments on original GPT2-Large** Following previous works (Garg et al., 2022; Chan et al., 2022a; Min et al., 2021), we train our model on a subset of equations and test ICL ability on the remaining unseen equations (using unseen ab pairs).This training process contributes to enhancing the transformer's comprehension of our instructions. The table below depicts the ICL results of an un-pretrained GPT2-Large model (indicating that it uses GPT2-Large weights directly without any training on our dataset). The results reveal that un-pretrained GPT2 struggles to comprehend our instructions and performs poorly in in-context learning.

Table 3: Experiments on original GPT2-Large

| ICL Accuracy | 0 ex | 1 ex | 3 ex | 7 ex | 15 ex | 23 ex |
|---|---|---|---|---|---|---|
| Info 0 nat | 0 | 0 | 0 | 0.014 | 0 | 0 |
| Info 1.02 nats | 0 | 0 | 7e-04 | 5.2e-05 | 0.001 | 0.001 |
| Info 3.22 nats | 0 | 1e-04 | 2e-04 | 8e-05 | 0 | 0 |
| Info 3.91 nats | 0.001 | 0.001 | 0.006 | 0.001 | 2e-04 | 2e-04 |
| Info 4.61 nats | 6e-04 | 9e-04 | 2.4e-05 | 6.7e-05 | 7e-04 | 0.007 |

**Double-side impact can also be observed in experiments on smaller transformers.** Table 4 illustrates the performance of a smaller 12-layer model, showing a similar transition on accuracy. However, compared to experiments using the larger 24-layer model, there's a noticeable decline in validation accuracy regardless of the task information provided. Even smaller models (e.g., 6 layers) perform poorly, with a validation accuracy of only 0.1258 despite precise task descriptions and 32 in-context examples. To thoroughly investigate how task information influences performance, we opt for the larger 24-layer transformer to capture performance variations more accurately.

Table 4: Ablation results given different amount of task info and 15 in-context examples.

| Task Info(nats) | 0 | 0.21 | 0.45 | 0.73 | 1.39 | 1.83 | 3.22 | 4.02 | 4.27 | 4.61 |
|---|---|---|---|---|---|---|---|---|---|---|
| Smaller Transformer(12 layers) | 0.1620 | 0.1034 | 0.1012 | 0.1004 | 0.1014 | 0.1014 | 0.1013 | 0.4507 | 0.771 | 0.9284 |
| Finetuned Vicuna-13b | 0.22 | 0.09 | 0.07 | 0.11 | 0.14 | 0.17 | 0.44 | 0.81 | 0.85 | 0.99 |

**Experiments on Vicuna-13b.** We replicate our experiments on larger models to verify that our experimental results were not due to a small model or poor understanding.

**(i) The attention hypothesis is still held on original Vicuna-13b.** We use the same prompt as the experiments using GPT-4 in Section 5.3. We ask Vicuna-13b to elucidate the mechanism underlying its output and calculate the proportion of instances where the model analyzes the in-context examples. We refer to this metric as "Example Analyzing Rate". The output accuracy and example analyzing rate are listed in Table 5. Similar to the pattern shown in Figure 3A, the example analyzing rate decreases with additional task description information. However, compared to GPT-4, the original Vicuna-13b model's relatively limited computing capabilities constrained its ability to accurately solve equations. Despite the increased attention focused on examples, the model faces challenges in learning the correct values of $a$ and $b$ from the in-context examples, leading to a rapid decline in

Table 5: Results on original Vicuna-13b.

| Task info (nats) | 0 | 0.11 | 0.21 | 0.45 | 0.71 | 1.39 | 2.41 | 3.22 | 3.91 | 4.61 |
|---|---|---|---|---|---|---|---|---|---|---|
| ICL Accuracy | 0.02 | 0.05 | 0.05 | 0 | 0.01 | 0 | 0.01 | 0.18 | 0.2 | 0.55 |
| Example Analyzing | 1 | 1 | 0.9 | 0.4 | 0.44 | 0.55 | 0.55 | 0.4 | 0.35 | 0.1 |

output accuracy. Even when provided with the exact task description, the original Vicuna-13b model may still struggle to follow the instructions and accurately calculate the result (only 0.55 accuracy).

**(ii) A similar negative impact of insufficient task description can also be observed in the case of the fine-tuned Vicuna-13b model.** We fine-tuned Vicuna-13b on our dataset for 3 epochs and results are given in Table 4. This fine-tuning improves the model's comprehension of our instructions and enhances its computational accuracy. Enhancements in ICL accuracy are noticeable across all task information settings, although the degree of improvement varies. Similar to experiments conducted on GPT-2, when low-level task information is added, the model may struggle to determine which source to prioritize for capturing the in-context ability. In such scenarios, the fine-tuning improvements in computational accuracy may provide limited assistance, as attention is diverted away from the examples.

**We conducted experiments on both smaller and larger models, confirming that our observations remain consistent across different model sizes.** Additional ablations on edge cases are detailed in Appendix E. These include scenarios where no task information is provided, or different amounts of task information are given without any in-context examples.

## 6 DISCUSSION AND LIMITATION

While we acknowledge the dual impact of task description and conduct some attention-based analysis, we don't provide a comprehensive explanation for the transition effect. In appendix F, we attempt to explain this using a simplified 1-layer transformer's attention mechanism. Understanding the transition phenomena in context learning is intriguing yet challenging; for instance, GROKKING (Power et al., 2022) only shows a sharp transition without clarifying it. Another work (Raventós et al., 2023) observed a similar transition, attributing it to a shift from the theoretical optimal task to the actual task. Our above study suggests that the transition in task description results from the conflict between task information and in-context examples, leading to model confusion during optimization. This switching between information sources causes the transition.

A potential limitation of this work lies in the synthetic experimental setting that has been employed to investigate the impact of task descriptions on in-context learning performance of Transformers. While this approach enables the systematic exploration of task description information and its influence on model performance, it may not fully capture the challenges encountered in real-world scenarios. Nevertheless, our method highlights the dual impact of task description on in-context learning, which is important to the community.

## 7 CONCLUSION

In conclusion, transformers have demonstrated remarkable performance across diverse applications, with in-context learning as a crucial technique. However, our grasp of its mechanisms remains limited. This study delves into the role of task descriptions in in-context learning for transformers, revealing their impact on performance. Our experiments in a synthetic environment underscore the importance of crafting task descriptions carefully to improve model performance and generalization, considering the impact transition. This study contributes to our understanding of in-context learning in transformers, paving the way for more effective real-world applications. Future research could focus on developing automated methods for generating optimal task descriptions to enhance model performance across tasks. Exploring learning mechanisms that seamlessly integrate task descriptions and examples would also be valuable.

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

## A    IMPLEMENTATION DETAILS

**Synthetic experiments.**    We use a standard decoder-only causal Transformer (Vaswani et al., 2017) with 24 layers, an embedding length of 256, and 8 attention heads. Following previous works (Garg et al., 2022; Chan et al., 2022a; Min et al., 2021), we train the model on our synthetic dataset for 200k steps to enhancing the transformer's comprehension of our instructions. The dataset is evenly split into training and testing sets, necessitating the model to grasp the instructions from the training set and perform in-context learning accordingly on the unseen equation tasks in the testing set. We conduct all experiments with a batch size of 128 on a single 3090 GPU, and use Adam optimizer with learning rate $1e^{-4}$. To reduce the randomness, we calculate the mean value of 3 runs, std indicated in Figure 1 and Figure 4.

**Experiments on CoFE.**    For experiments on the natural language task CoFE (An et al., 2023), we follow their approach and use pretrained GPT2-Large as our model. All experiments conducted with a batch size of 4 on a single 3090 GPU. Similarly, we fine-tune model on a subset of data and test in-context learning ability on unseen language combinations, using Adam optimizer with learning rate $1e^{-4}$ and calculate the mean value of 3 runs. Std indicated in Figure 5.

**Experiments with Vicuna-13b.**    Both the original and fine-tuned experiments with Vicuna-13b utilize pretrained model weights (Chiang et al., 2023). The training and testing sets used in previous GPT2 experiments are also employed here. Vicuna-13b is fine-tuned on the training set for 3 epochs, while other settings remain at their default values.

## B    THE DERIVATION OF EQUATION 3

In the following two sections, we have primarily drawn upon the proofs VAE (Kingma & Welling, 2013) and in Belghazi et al. (2018) as key literature sources to prove our claims.

Using Bayes rule, we can obtain the following derivation:

$$
\begin{aligned}
&\mathbb{KL}(q_\theta(t|d,c,q)|p(t|r,d,c,q)) \\
&= \mathbb{E}_{q_\theta(t|d,c,q)}[\log q_\theta(t|d,c,q) - \log p(t|r,d,c,q)] \\
&= \mathbb{E}_{q_\theta(t|d,c,q)}\left[\log q_\theta(t|d,c,q) - \log \frac{p(r|t,d,c,q)p(t|d,c,q)}{p(r|d,c,q)}\right] \\
&= \mathbb{E}_{q_\theta(t|d,c,q)}\left[\log \frac{q_\theta(t|d,c,q)}{p(t|d,c,q)} - \log p(r|t,d,c,q)\right] + \log p(r|d,c,q) \\
&= \mathbb{KL}(q_\theta(t|d,c,q)|p(t|d,c,q)) - \mathbb{E}_{q_\theta(t|d,c,q)}\log p(r|t,d,c,q) + \log p(r|d,c,q)
\end{aligned}
\tag{9}
$$

## C    THE DERIVATION OF EQUATION 4

We can rewrite the reconstruction error with the conditional distribution $p(t|d) = p(t,d)/p(d)$:

$$
\begin{aligned}
\mathcal{R} &= \mathbb{E}_{q(t)}\mathbb{E}_{q(d|t)}[-\log p(t|d)] = \mathbb{E}_{q(t,d)}\left[\log \frac{q(t,d)}{p(t,d)}\right] - \mathbb{E}_{q(t,d)}[\log q(t,d)] + \mathbb{E}_{q(d)}[\log p(d)] \\
&= \mathbb{KL}(q(t,d)|p(t,d)) - \mathbb{E}_{q(t,d)}[\log q(t,d)] + \mathbb{E}_{q(d)}[\log p(d)],
\end{aligned}
\tag{10}
$$

where the first term is KL divergence, the second term is the joint entropy $H_q(t,d)$. We focus on the third term:

$$
\mathbb{E}_{q(d)}[\log p(d)] = \mathbb{E}_{q(d)}[\log \frac{p(d)}{q(d)}] + \mathbb{E}_{q(d)}[\log q(d)] = -\mathbb{KL}(q(d)|p(d)) + H_q(d)
\tag{11}
$$

We bring Eq. 11 into Eq. 10, then the joint entropy and entropy can be formulated as:

$$
\begin{aligned}
-\mathbb{E}_{q(t,d)}[\log q(t,d)] + H_q(d) &= -\mathbb{E}_{q(t,d)}\left[\log \frac{q(t,d)}{q(t)q(d)}\right] + \mathbb{E}_{q(t)}[\log q(t)] \\
&= -I_q(t;d) + H_q(t)
\end{aligned}
\tag{12}
$$

Table 6: Examples of data in CoFE.

| Category | In-context Examples | Test Case |
|---|---|---|
| Primitive Substitution | input:shark
output:NONE(SHARK,NONE,NONE)
input:A girl grew the boy.
output:DRAW(Girl,BOY,NONE) | input:The shark drew a boy.
output:DRAW(SHARK,BOY,NONE) |
| Primitive Structural Alternation | input:The goose baked.
output:BAKE(GOOSE,NONE,NONE)
input:A teacher noticed a chicken.
output:NOTICE(TEACHER,CHICKEN,NONE) | input:A teacher baked the chicken.
output:BAKE(TEACHER,CHICKEN,NONE) |
| Phrase Recombination | input:Logan mailed Stella the cake in the pile.
output:MAIL(LOGAN,IN,STELLA)
input: The goose rolled a baby in a room.
output:ROLL(GOOSE,IN,NONE) | input:A visitor in the pile rolled a resident.
output:ROLL(IN,RESIDENT,NONE) |

Since the KL-divergence is non-negative, we obtain the bound:

$$\begin{aligned}
\mathcal{R} &= \mathbb{KL}(q(t,d)|p(t,d)) - \mathbb{KL}(q(d)|p(d)) - I_q(t;d) + H_q(t) \\
&\leq \mathbb{KL}(q(t,d)|p(t,d)) - I_q(t;d) + H_q(t)
\end{aligned} \tag{13}$$

## D    COFE DATASET

CoFE dataset (An et al., 2023) is constructed based on COGS, a compositional generalization benchmark designed for the fine-tuning paradigm. Here, compositional generalization refers to understanding and producing novel expressions by recombining known components in language, and is an important human ability. COGS, as well as CoFE, are designed for semantic parsing tasks. In these datasets, the training set covers all the primitives but lacks certain combinations, and the test set is made up of these missing combinations, so the model has to learn to translate natural language expressions into semantic representations.

**Why we experiment on CoFE dataset.**    We choose CoFE for our experiments in Section 5.4 for two primary reasons: 1) CoFE is an NLP dataset designed to tackle more complicated and realistic tasks. We conduct additional experiments to validate that insights gained from synthetic experiments also apply to real-world tasks. 2) The task description in CoFE can be partially provided, allowing us to adjust the amount of information given to transformers. This enables us to study the impact of task information on in-context learning.

**Explanation for CoFE dataset.**    CoFE is a NLP incontext-learning datset based on compositional generalization tasks. Compositional generalization refers to the ability of a model to generalize its understanding and make predictions about novel combinations of components based on its training data. In other words, it's the capacity to understand and perform tasks involving new compositions of elements or concepts that it hasn't explicitly encountered during training.

For example, consider a language model trained on sentences like "The cat is on the mat" and "The dog is in the garden." If the model has good compositional generalization, it should be able to understand and generate correct responses to novel sentences like "The cat is in the garden" or "The dog is on the mat" even though it hasn't seen those exact combinations of words during training.

CoFE requires the model to perform compositional generalization on semantic parsing tasks. The objective involves predicting semantic representations of input sentences, such as subjects and objects. However, the queries provided are recombinations of the in-context examples, compelling the transformer to grasp grammar fully and predict on new compositions of elements or concepts not explicitly encountered in the examples.

Thus, the task type in CoFE can be determined by two factors: the type of recombination (concerning the query) and the type of semantic representation to be predicted (concerning the output). Some data examples are given in Table 6.

**How we use CoFE.**    In our experiments, we employ 3 types of recombination to predict 4 types of semantic representations, resulting in a total of 12 different tasks. We randomly select 4 of these tasks

for pre-training the model to ensure it comprehends our instructions (ensuring coverage of all types of semantic representations in the training set), while the remaining 8 are reserved for the in-context learning test. We imply the type of recombination and the type of semantic representation in task description. And in Part Task Info experiment, we only imply the type of recombination, leaving the model to learn which type of semantic representation it should predict based on the examples.

We experiment on all three info settings under different numbers of in-context examples. In task description, the combination categories are tokenized as 1,2,3 and the target primitive type are denoted as 11-14. All words in CoFE are tokenized starting from 100 to avoid messing up with task description.

**The relation between experiments on CoFE and experiments on synthetic tasks: The conclusions of synthetic experiments are still held.** On CoFE, we evaluated model's ICL ability given Full Task Info, Part Task Info, No Task Info, and whether asked to predict the real task or not. Still, using task prediction as proxy task can significantly improve accuracy, while a negative impact is observed when the task description is insufficient.

# E    ADDITIONAL ABLATIONS

## E.1    ABLATION: NO TASK INFORMATION DURING TRAINING

We present the transformer's accuracy given no task information and different number of in-context examples. It can be depicted in Table 7 that the accuracy grows with in-context example number. This table actually refers to zero mutual information in Figure 1A.

## E.2    ABLATION: NO IN-CONTEXT EXAMPLES DURING TRAINING

Table 8 lists the transformer's accuracy given different amount of task information and no in-context examples. When given maximal info (4.605 nats, referring to totally accurate task description), the transformer performs best. This indicates transformer's ability in understanding well-designed task description. Also, it can be seen that under no in-context example setting, the accuracy grows with task information gain. The growing trend speeds up when more task information added (around 3.2 nats, which is close to the transition threshold). Such performance pattern aligns with experiments given both task description and in-context examples.

Table 7: Given different number of in-context examples, no task information provided.

| Number of In-context Examples | 0 | 1 | 3 | 5 | 7 | 9 | 11 | 15 | 17 | 23 |
|---|---|---|---|---|---|---|---|---|---|---|
| Accuracy | 0.1017 | 0.1117 | 0.1198 | 0.1320 | 0.2093 | 0.1875 | 0.2955 | 0.3670 | 0.4267 | 0.5367 |

Table 8: Given different amount of task information, no in-context examples provided.

| Task Info (nats) | 0 | 0.104 | 0.207 | 0.223 | 0.329 | 0.445 | 0.713 | 1.022 | 1.386 | 1.833 | 2.303 | 2.996 | 3.219 | 3.307 | 3.506 | 3.624 | 3.759 | 3.912 | 4.094 | 4.317 | 4.6052 |
|---|---|---|---|---|---|---|---|---|---|---|---|---|---|---|---|---|---|---|---|---|---|
| Accuracy | 0.1017 | 0.1024 | 0.1055 | 0.1002 | 0.1056 | 0.1035 | 0.1040 | 0.1053 | 0.1013 | 0.1100 | 0.1210 | 0.1189 | 0.1248 | 0.1574 | 0.1555 | 0.2404 | 0.2572 | 0.2834 | 0.5225 | 0.7857 | 0.9107 |

# F    EXPLANATION FOR EXPERIMENTAL PHENOMENON

**We explore the attention mechanism behind the phenomenon based on a 1-layer position-encoding-free transformer.** We follow Tian et al. (Tian et al., 2023) to construct a simplified 1-layer position-encoding-free transformer, to understand how transformers work in in-context learning, especially the attention mechanisms in learning from task description and in-context examples. This simplified model can elucidate the interference between insufficient information and inadequate in-context examples in attention, as well as illustrate how a low level of task information content affects in-context learning.

## F.1    PROBLEM SETTING

We follow Tian et al. (Tian et al., 2023) to construct a simplified 1-layer transformer, which contains one softmax self-attention layer followed by one decoder layer which predicts the next token. The

analysis is conducted under the following assumptions: no positional encoding; long input sequence; the decoder layer learns much faster than the self-attention layer.

## F.2 NOTATIONS

Given input sequence $X = [x_1, x_2, \ldots, x_T]$, the task of the transformer is to predict the next token $x_{T+1}$. We call $x_T = m$ as the **query token** of the sequence, and $x_{T+1} = n$ as the **next token** to be predicted. Other tokens $x_t (1 \leq t \leq T - 1)$ are called **contextual tokens**. In our experiments, contextual tokens can be split into task description $d$ and in-context examples $e$. The other notations are listed in Table 9.

Table 9: Notations.

| | |
|---|---|
| $n$ | Representation of next token in formulation |
| $l, l'$ | Representation of distinct contextual tokens in formulation |
| $d$ | Task description token |
| $e$ | In-context example token |
| $P(l\|n)$ | Conditional probability of contextual token $l$ given certain query token and next token to be predicted as n |
| $r_{l/l'\|n}(t)$ | Relative gain between $l$ and $l'$ for next token $n$ |
| $c_{ln}$ | Un-normalized attention score given next token $n$ |
| $R_a, R_b$ | Possible range of a and b given in task description |
| $n_e$ | Number of given in-context examples |
| $S_e$ | Total number of all possible example pairs |
| $z_m$ | Attention logits for a query token m |
| $\dot{z_m}$ | Dynamics of self-attention |
| $n \in \Psi^{-1}(m)$ | All next token $n$ that can be predicted by the query token $m$ ($P(n\|m) > 0$) |
| $f_n$ | $l_2$-norm attention score corresponding to the position of $n$ |

## F.3 THEOREMS

**Theorem 1.** Under certain simplifications, we simplified the conditional probability as $P(d|n) = 1/R_a R_b$ and $P(e|n) = n_e/S_e$. Then the relative attention gain between task description $d$ and in-context examples $e$ can be written as:

$$r_{d/e|n}(t) = (\frac{S_e}{R_a R_b n_e})^2 - 1$$

This formulation indicates that the difference between $P(d|n)$ and $P(e|n)$ can decide the distribution of self-attention, resulting in two possible scenarios:

1) If either $d$ or $e$ exhibits decisive certainty, the relative gain will be significant, prompting the self-attention mechanism to concentrate on learning from this particular type of information.

2) If neither $d$ nor $e$ can decisively outweigh the other, leading to both $|r_{d/e|n}(t)|$ and $|r_{e/d|n}(t)|$ being close to 1, the attention mechanism lacks emphasis and may distribute randomly. This can hinder effective learning.

**Theorem 2.** Given Lemma 4 in Tian et al. (Tian et al., 2023) ($n'$ is a possible next token different from $n$):

$$\dot{z_m} = \eta_Z \gamma \sum_{n \in \Psi^{-1}(m)} diag(f_n) \sum_{n' \neq n} \beta_{nn'} (f_n f_n^T) - I) f_{n'}$$

We neglect learning coefficients $\eta_Z, \gamma, \beta_{nn'}$ and assume the transformer is given adequate in-context examples. Under certain simplifications, the dynamics of self-attention can be formulated as:

$$\dot{z_m} = \eta_Z \gamma \sum_{n \in \Psi^{-1}(m)} \sum_{n' \neq n} \beta_{nn'} [\frac{1 - R_a^2 R_b^2}{R_a^3 R_b^3}, \frac{n_e^2}{R_a R_b S_e^2}]$$

The second term of the above formula related to the corresponding attention learning speed of in-context examples, which diminishes with insufficient task description (indicated by larger values of $R_a$ and $R_b$ as the task information becomes more ambiguous). This term remains unaffected only when the task description is precise ($R_a = R_b = 1$).

### F.4 PROOF STRETCH

We assume the task description and in-context examples as single, distinct tokens for simplicity. As the query can be randomly selected given task description and in-context examples, the probabilities $P(m|d)$ and $P(m|e)$ are neglected for simplicity. And we approximate the conditional probability $P(d|n)$ and $P(e|n)$ , solely to elucidate the relationship between the task description and in-context examples. Then substitute all these terms into Theorem 3 from Tian et al. (Tian et al., 2023), yielding Theorem 1 here.

To study the impact of insufficient task description on learning, we assume the transformer is given adequate in-context examples that given a certain query, specific in-context examples will reliably predict specific next token $n$. On the contrary, task description is insufficient so there exists $n'$ that $P(d|n') \neq 0, P(d|n) \neq 0$. Substitute the above terms into Lemma 4 in Tian et al. (Tian et al., 2023) and drop non-essential constant terms, the dynamics of self-attention can be formulated as Theorem 2 here.

### F.5 DETAILED PROOF FOR THEOREM 1

According to Tian et al. (Tian et al., 2023), for a next token n and its two distinct tokens $l$ and $l'$, the dynamics of the relative self-attention gain can be formulated as:

$$r_{l/l'|n}(t) = c_{ln}^2(t)/c_{l'n}^2(t) - 1$$

Here "distinct tokens" refers to contextual tokens which appear only for a single next token (given certain query, the next token n can only be predicted by this distinct token). And $c_{ln}$ refers to un-normalized attention score given next token n.

For simplicity, assume the task description and in-context examples as single, distinct tokens, and denoted as 'd' and 'e'. As the query can be randomly selected given task description d and in-context examples e, the probabilities $P(m|d)$ and $P(m|e)$ are neglected for simplicity. Then the dynamics of the relative self-attention gain between task description and examples can be formulated as:

$$r_{d/e|n}(t) = C \cdot P(d|n)^2(t)/P(e|n)^2(t) - 1$$

Under our experiment setting, the probabilities can be simplified as:

$$P(d|n) = 1/R_a R_b, P(e|n) = n_e/S_e$$

Here $R_a, R_b$ denote the possible range of a and b given in task description, and $n_e$ denotes number of given in-context examples, $S_e$ denotes the total number of all possible example pairs (assume $S_e = 100$ for following analysis for easier calculation).

In this ideal situation (non-essential constant term neglected), the relative attention gain can be written as:

$$r_{d/e|n}(t) = (\frac{S_e}{R_a R_b n_e})^2 - 1$$

This formulation indicates that the difference between $P(d|n)$ and $P(e|n)$ can decide the distribution of self-attention. And there are two possible scenarios.

First, if either one of d or e has decisive certainty, the relative gain will be high enough to concentrate self-attention, so that the transformer can focus on learning from this certain kind of information.

For example, given exact task description ($R_a = R_b = 1$), then the relation gain is $r_{d/e|n}(t) = 99$ and $r_{e/d|n}(t) = -0.99$. Or given very ambiguous task description ($R_a = R_b = 10$) and adequate examples ($n_e = 10$), the relation gain is $r_{d/e|n}(t) = -0.99$ and $r_{e/d|n}(t) = 99$. The significant difference ensures that the transformation focuses attention on specific parts, learning from a more effective source of information.

However, here comes the second scenario if neither d nor e can decisively outweighs the other, resulting in $|r_{d/e|n}(t)|$ and $|r_{e/d|n}(t)|$ both near 1. Then the attention has no emphasis and can distribute randomly, which harms effective learning.

As a case of example, assume $R_a = R_b = 3$ and $n_e = 10$, then $r_{d/e|n}(t) = 0.23$ and $r_{e/d|n}(t) = -0.19$, resulting in no significant relative attention gain.

From this simplified model, it can be inferred that insufficient information and inadequate in-context examples interferes with each other in attention, while task description with abundant information can aid self-attention concentration.

This also agrees with our experiment in attention ratio in Fig 4A. Significant attention ratio change can only be witnessed when task info is very low (around 0) or very high (beyond 4). The attention distribution remains unchanged when given insufficient task description, inferring that insufficient task info does little help in self-attention.

### F.6 Detailed proof for Theorem 2

Lemma 4 in Tian et al. (Tian et al., 2023) gives the formulation of the dynamics of self-attention ($\dot{z_m}$):

$$\dot{z_m} = \eta_Z \gamma \sum_{n \in \Psi^{-1}(m)} diag(f_n) \sum_{n' \neq n} \beta_{nn'} (f_n f_n^T) - I) f_{n'}$$

$\eta_Z, \gamma, \beta_{nn'}$ correspond to learning coefficients and are neglected for simplicity in following analysis. $n \in \Psi^{-1}(m)$ refers to all next token "n" that can be predicted by the present token "m" ($P(n|m) > 0$), and $n'$ is a possible next token different from "n". $f_n$ denotes $l_2$-norm attention score corresponding to the position of "n", and can be simplified here as $[P(l_1|n), \cdots, P(l_1'|n)]^T$.

To study the impact of insufficient task description on learning, assume the transformer is given adequate in-context examples that given a certain query, $P(e|n) = n_e/S_e$ while $P(e|n') = 0$ ($n' \neq n$, and e refers to a combination of examples). On the contrary, task description "d" is insufficient so there exists $n'$ that $P(d|n') = P(d|n) = 1/R_a R_b$. Under the above simplifying assumptions, we have:

$$f_n = [\frac{1}{R_a R_b}, \frac{n_e}{S_e}]^T, f_{n'} = [\frac{1}{R_a R_b}, 0]^T$$

Substitute the above formula and drop non-essential constant terms, the dynamics of self-attention can be formulated as:

$$\dot{z_m} = \eta_Z \gamma \sum_{n \in \Psi^{-1}(m)} \sum_{n' \neq n} \beta_{nn'} [\frac{1 - R_a^2 R_b^2}{R_a^3 R_b^3}, \frac{n_e^2}{R_a R_b S_e^2}]$$

The second term of the above formula related to the corresponding attention learning speed of in-context examples, which degrades with insufficient task description (the more ambiguous task info given, the larger $R_a, R_b$). Only when the task description is accurate ($R_a = R_b = 1$) can this term be unaffected.

The aforementioned deduction can somewhat reveal how insufficient task description impact the attention learning speed of in-context examples. The resulting negative impact can slow down the learning process and may even harm final results.

