# OpenReview forum: "Insufficient Task Description can Impair In-context Learning: A Study from Information Perspective"
_ICLR.cc/2025/Conference — ICLR 2025 Conference Withdrawn Submission_

### Official Review · Reviewer_14bA · 2024-11-02

**Soundness:** 3
**Presentation:** 3
**Contribution:** 2
**Rating:** 6
**Confidence:** 3

**Summary:**

This paper seeks to investigate how task descriptions, and their quality, affect in-context learning performance in transformers. In particular, the authors explore how insufficiently detailed task descriptions can lead to lower performance, through synthetically generated task descriptions as a way to control for the level of detail provided by the task description. Empirical evaluations demonstrate that while vague descriptions impair performance, more informative descriptions significantly enhance in-context learning.

**Strengths:**

The strengths of the paper are as follows:
- The paper is well written and provides a timely study on a less-explored aspect of in-context learning.
- The resulting insights provide a wide range of applicability across large language models.
- The synthetic experimental setup further provides a simple setting for future work to leverage.

**Weaknesses:**

The reviewer's primary concern with this paper is that the analysis heavily relies on the synthetic tasks which may not accurately reflect real-world applications.

**Questions:**

The reviewer wonders if the authors could help provide some additional intuitive explanations on why insufficient descriptions degrade performance in cases where in-context examples should theoretically compensate.

---

### Official Review · Reviewer_QtBu · 2024-11-02

**Soundness:** 2
**Presentation:** 1
**Contribution:** 1
**Rating:** 1
**Confidence:** 5

**Summary:**

This paper investigates the impact of task descriptions on the in-context learning performance of transformers. Through experiments on both synthetic and real-world tasks, the study demonstrates that insufficient task descriptions can harm performance even when a sufficient number of in-context examples are provided. Conversely, either complete task descriptions or a sufficient number of in-context examples without task descriptions can achieve relatively high performance. The study highlights the critical role of task descriptions in the in-context learning of transformers.

**Strengths:**

1. The focus on the role of task descriptions in transformer in-context learning is novel.
2. Experiments were conducted on both synthetic and real-world datasets, testing models from self-trained transformers to language models like GPT-2 Large and Vicuna-13B.

**Weaknesses:**

1. The most severe problem is the **unreasonable theorem** in Section 3.
    - **Flawed Equation 1.** The correct objective to maximize the log-likelihood of the transformer $q_\theta(r|d,c,q)$ using the given data should be
    $
    E_{p(d,c,q)} E_{p(r|d,c,q)} [\log q_\theta(r|d,c,q)].
    $
    However, Equation 1 states
    $
    E_{p(d,c,q)} E_{q_{\theta}(r|d,c,q)} [\log p(r|d,c,q)].
    $
    This term is definitely not the log-likelihood of the data as it mistakenly utilizes the ground truth distribution $p(r|d,c,q)$ within the objective function. I suggest that the authors provide a step-by-step derivation of their objective function, explaining their reasoning at each step.
    - **Problematic interpretation of Equation 3.** The authors derive Equation 3 and state that the KL divergence term on the right side contributes to maximizing log-likelihood. However, unlike the typical Evidence Lower Bound (ELBO), where the left side includes an optimizable distribution, here, $\log p(r|d,c,q)$ on the left of Equation 3 is a **fixed** ground truth distribution. **This raises the question**: how does optimizing the KL divergence term on the right improve a **fixed** distribution on the left? I recommend the authors elaborate on this interpretation and explicitly connect the logic from Equation 3 to their claims. In addition, I suggest the authors compare Equation 3 with the standard ELBO and discuss any key differences.
    - **An intuitive example of logical flaws.** I use one example to highlight the theorem's absurdity after all the logical flaws have accumulated. If we replace the variable $t$ with a random variable $z$ indicating "whether tomorrow will be sunny in my hometown," and use the same logits from Equation 1 to 3, we absurdly conclude that forecasting the weather in my hometown contributes to the log-likelihood maximization for their transformers.  This contradiction stems from the accumulated logical flaws, including the previous two. I thus suggest the authors carefully examine their theorem and justify why the task label ($t$) prediction is meaningful in the context of their problem.

2. **Inappropriate Experimental Design**: In the real-world experiments, some "insufficient" task descriptions are actually incorrect. For instance, in the spelling task, the full task info is "extract the second letter of the input word," while the partial task info is "extract letter of the input word." The latter implies extracting each letter, which is a different task. Ideally, answers given the full task info should be a subset of those given the partial task info. A more appropriate partial task description would be "extract a certain letter of the input word" in the previous case. I suggest that the authors revise their partial task descriptions to ensure they are truly subsets of the full task descriptions. Additionally, I recommend them discuss their criteria for determining "insufficient" task descriptions and analyze how their choices might impact their results and conclusions.

**Questions:**

1.What is the precise definition of the “attention ratio” in Figure 3? Could the authors provide exact formulas for its calculation?

---

### Official Review · Reviewer_9WWH · 2024-11-03

**Soundness:** 1
**Presentation:** 2
**Contribution:** 2
**Rating:** 3
**Confidence:** 4

**Summary:**

The paper investigates how the task description information affects the performance of in-context learning in transformers. It introduces a synthetic dataset based on modular arithmetic equations to examine these effects under controlled conditions. The study finds that insufficient task description can impair performance, while sufficiently informative descriptions significantly improve model accuracy. The authors conduct additional experiments on a synthetic dataset and a real dataset (CoFE dataset), concluding that their insights can generalize across contexts.

In my point of view, the paper analyzes a well-known phenomenon in in-context learning, providing theoretical insights and experiments on a highly restrictive synthetic dataset that does not generalize easily to real-world tasks. Consequently, the conclusions remain limited in practical applicability and reaffirm what is already intuitively understood about task description information in in-context learning.

**Strengths:**

- The paper studies an important aspect of in-context learning, analyzing the contribution of task description in in-context learning.
- The paper tries to provide a theoretical understanding, which is not satisfactory but necessary.

**Weaknesses:**

- Limited Practical Impact of Theoretical Formulation: Section 3 introduces a theoretical framework with equations involving KL divergence and mutual information to motivate the role of task descriptions. However, these equations, especially Equation (5), are not integrated into the experiments and thus do not guide the empirical work in any meaningful way. The formulation could be streamlined or more directly connected to the paper’s practical findings.

- Restricted Applicability of Mutual Information Calculation (Equation 6): The authors quantify task description information in Equation (6) by defining bounds on task parameters. This works well for their synthetic dataset, where parameters are precisely controlled. However, the method lacks applicability to real-world datasets, where task definitions are more complex and less structured. The authors do not demonstrate how to extend this metric to real datasets like CoFE, which limits the study’s relevance beyond synthetic setups. Neither the theoretical analysis nor the form of synthetic data make sense to CoFE.

- Limited Generalizability of Synthetic Data to Real-World Tasks: The synthetic data structure, based on simple arithmetic equations, does not represent the complexity found in most real-world datasets. Real tasks, such as those in natural language processing, often require interpreting nuanced instructions rather than solving modular arithmetic problems. Consequently, the insights gained from these synthetic tasks may not fully transfer to more realistic settings.

- Unclear Notations and Words: The paper contains many abbreviations and notations for readers to guess. For example, no ex, 1 ex, 3 ex in Figure 1. Hq(t) in equation (5). Not Pred Task in Figure 5.

**Questions:**

- Section 3 presents several equations related to mutual information and KL divergence, yet these are not subsequently used to guide or interpret the experimental results. Could you clarify the intended purpose of these equations in relation to your experiments? How might they theoretically inform practical findings?

- In Equation (6), you define mutual information based on ranges for synthetic parameters a and b. Do you envision a way to calculate or estimate mutual information in real datasets where task descriptions lack discrete bounds? How might this approach extend to datasets like CoFE, where task descriptions are less structured?

- Given the highly structured nature of the synthetic data, how do you envision your findings scaling to real-world datasets that are more complex and less deterministic? Are there specific domains beyond modular arithmetic where you believe your method would be particularly applicable?

---

### Note · Authors · 2024-12-18

I have read and agree with the venue's withdrawal policy on behalf of myself and my co-authors.